# Acquisition of T6SS Effector TseL Contributes to the Emerging of Novel Epidemic Strains of *Pseudomonas aeruginosa*

Anmin Ren,[a] Minlu Jia,[a] Jihong Liu,[c] Tian Zhou,[a] Liwen Wu,[b] (ID)Tao Dong,[b] Zhao Cai,[a] Jiuxin Qu,[f] Yang Liu,[c] (ID)Liang Yang,[a,d,e] (ID)Yingdan Zhang[a]

[a]School of Medicine, Southern University of Science and Technology, Shenzhen, Guangdong, People's Republic of China
[b]School of Life Sciences, Southern University of Science and Technology, Shenzhen, Guangdong, People's Republic of China
[c]Medical Research Center, Southern University of Science and Technology Hospital, Shenzhen, Guangdong, People's Republic of China
[d]Shenzhen Third People's Hospital, The Second Affiliated Hospital of Southern University of Science and Technology, National Clinical Research Center for Infectious Disease, Shenzhen, Guangdong, People's Republic of China
[e]Key University Laboratory of Metabolism and Health of Guangdong, Southern University of Science and Technology, Shenzhen, Guangdong, People's Republic of China
[f]Shenzhen Third People's Hospital, The Second Affiliated Hospital of Southern University of Science and Technology, Shenzhen, Guangdong, People's Republic of China

Anmin Ren and Minlu Jia contributed equally to this work. Author order was determined by drawing straws.

**ABSTRACT** *Pseudomonas aeruginosa* is an opportunistic pathogen with multiple strategies to interact with other microbes and host cells, gaining fitness in complicated infection sites. The contact-dependent type VI secretion system (T6SS) is one critical secretion apparatus involved in both interbacterial competition and pathogenesis. To date, only limited numbers of T6SS-effectors have been clearly characterized in *P. aeruginosa* laboratory strains, and the importance of T6SS diversity in the evolution of clinical *P. aeruginosa* remains unclear. Recently, we characterized a *P. aeruginosa* clinical strain LYSZa7 from a COVID-19 patient, which adopted complex genetic adaptations toward chronic infections. Bioinformatic analysis has revealed a putative type VI secretion system (T6SS) dependent lipase effector in LYSZa7, which is a homologue of TseL in *Vibrio cholerae* and is widely distributed in pathogens. We experimentally validated that this TseL homologue belongs to the Tle2, a subfamily of T6SS-lipase effectors; thereby, we name this effector TseL (TseL[PA] in this work). Further, we showed the lipase-dependent bacterial toxicity of TseL[PA], which primarily targets bacterial periplasm. The toxicity of TseL[PA] can be neutralized by two immunity proteins, TsiP1 and TsiP2, which are encoded upstream of *tseL*. In addition, we proved this TseL[PA] contributes to bacterial pathogenesis by promoting bacterial internalization into host cells. Our study suggests that clinical bacterial strains employ a diversified group of T6SS effectors for interbacterial competition and might contribute to emerging of new epidemic clonal lineages.

**IMPORTANCE** *Pseudomonas aeruginosa* is one predominant pathogen that causes hospital-acquired infections and is one of the commonest coinfecting bacteria in immunocompromised patients and chronic wounds. This bacterium harbors a diverse accessory genome with a high frequency of gene recombination, rendering its population highly heterogeneous. Numerous *Pa* lineages coexist in the biofilm, where successful epidemic clonal lineage or strain-specific type commonly acquires genes to increase its fitness over the other organisms. Current studies of *Pa* genomic diversity commonly focused on antibiotic resistant genes and novel phages, overlooking the contribution of type VI secretion system (T6SS). We characterized a *Pa* clinical strain LYSZa7 from a COVID-19 patient, which adopted complex genetic adaptations toward chronic infections. We report, in this study, a novel T6SS-lipase effector that is broadly distributed in *Pa* clinical isolates and other predominant pathogens. The study suggests that hospital transmission may raise the emergence of new epidemic clonal lineages with specified T6SS effectors.

Address correspondence to Liang Yang, yangl@sustech.edu.cn, or Yingdan Zhang, zhangyd6@sustech.edu.cn.

The authors declare no conflict of interest.

**KEYWORDS** *Pseudomonas aeruginosa*, T6SS effector, TseL, pathogenesis

*P*seudomonas aeruginosa* (*Pa*) is a predominant opportunistic Gram-negative patho-gen that causes a wide range of infections with high morbidity and mortality, especially for patients who are suffering from immunocompromised conditions (1). The success of *Pa* as a nosocomial pathogen is largely attributed to its versatile viru-lence mechanisms and metabolic activities (2). Moreover, *Pa* usually forms biofilms on infection sites and/or clinical devices, where cells are nested in a biofilm matrix with high population density (3). *Pa*, in the biofilm communities, can interact with biotic and abiotic environments to survive and gain fitness in the complicated infection sites, rendering biofilm-associated infections difficult to eradicate (4–6).

*Pa* commonly interacts with neighboring organisms in a variety of ways, in which the contact-dependent type VI secretion system (T6SS) is involved in fitness competi-tion and pathogenesis (7). T6SS is broadly distributed in over 25% Gram-negative bac-teria, which delivers various effectors to interact with bacterial biotic and/or abiotic environment (8, 9). So far, *Pa* is shown to encode three classes of T6SSs, namely, H1-, H2-, and H3-T6SS, which delivers a considerable number of effectors (10). The contribu-tion of *Pa* T6SS in competition and/or in pathogenesis is mainly decided by the activ-ities of its secreted effectors. To date, a series of effectors has been predicted by com-parative genomic analysis in *Pa* clinical strains. However, only a limited number of effectors have been validated in *Pa* model organisms (e.g., PAO1 and PA14) (11–13). More recently, effector combinations are revealed to contribute to intraspecific diver-sity of *Pa* T6SS as well as the associated bacterial pathogenesis (13).

The population structure of *Pa* is well documented with panmictic-epidemic nature (14). The *Pa* harbors a diverse accessory genome and shows high frequency of gene recombination across random loci (14, 15). Numerous *Pa* lineages coexist in the biofilm, where successful epidemic clonal lineage or strain-specific type commonly acquires genes to increase its fitness over the same species (1, 16, 17). Over the past decades, investigation of gene acquisition by *Pa* mainly focuses on the aspects of antibiotic resistance, pathoge-nicity and those could counterbalance the fitness cost in a specific niche. A recent work reported the intraspecific diversity of accessory effectors in *Pa* local population (13). However, it remains less illustrated whether *Pa* acquires diverse T6SS effector loci to shape epidemic strains against environmental stresses. Investigation of diverse T6SS effectors from *Pa* clinical isolates will shed light on its strain-level variation of *Pa* pathogenesis as well as novel biomarkers of diagnosis.

In a recent study, we characterized a *P. aeruginosa* clinical strain LYSZa7 from a COVID-19 patient, which adopted complex genetic adaptations toward chronic infections. A series of T6SS related genes have been annotated in LYSZa7 with a homologue of well recognized lipase-effector protein TseL in *V. cholerae*, which belongs to the T6SS lipase effectors (Tle) that hydrolyze phosphodiester bonds of target cells. The TseL in *V. cholerae* belongs to the Tle2 superfamily of phospholipase (18, 19), which was predicted in a *P. aeruginosa* clinical iso-late PA7 by informatic analysis with functions undemonstrated. In this study, we proved that this TseL homologue is a Tle2 lipase-effector protein (termed as TseL) with two immunities (termed as TsiP1 and TsiP2), and it is ubiquitous in 104-negative pathogens and exists in a large set of *P. aeruginosa* clinical isolates. The TseL in *P. aeruginosa* (TseL[PA]) targets bacterial periplasm, causing membrane damage and contributing to both intra- and interspecies com-petition. Also, TseL[PA] is shown to play critical roles in host invasion and resistance toward phagocytosis.

## RESULTS

**Identification of a novel T6SS effector in *P. aeruginosa* clinical isolate.** When characterizing the genome of the *P. aeruginosa* LYSZa7 strain that was isolated from a COVID-19 patient (20), we found a putative VgrG-dependent lipase that does not exist in the PAO1 and PA14 model strains (WP_200200543.1). This putative lipase encoding gene LYSZa7_12680 is adjacent to a homolog of *vgrG* (LYSZa7_12660) with two predicted

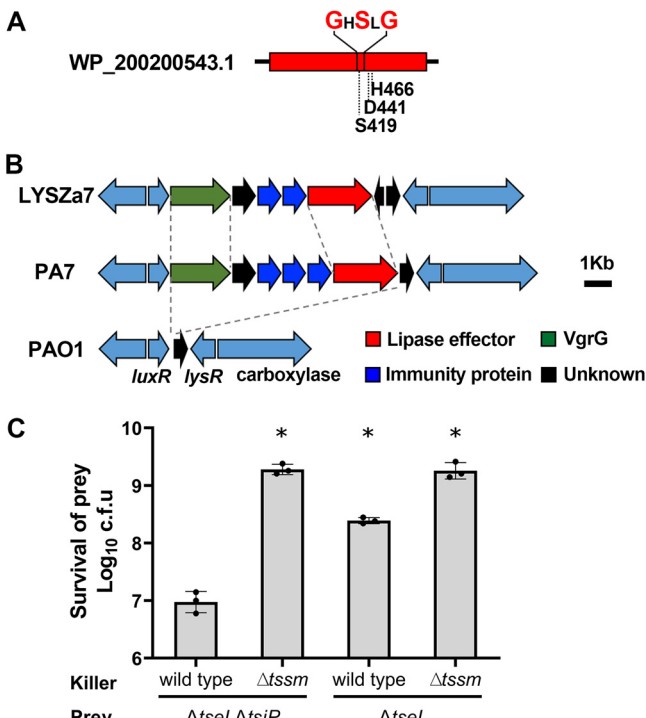

**FIG 1** Illustration of the (A) conserved motifs of VgrG-dependent lipase (WP_200200543.1) in *P. aeruginosa* LYSZa7 and (B) the comparison of this VgrG-lipase-immunity protein cassette of LYSZa7 with those of PA7 and PAO1. (C) competition assay of LYSZa7 wild-type and the T6SS-defective mutant LYSZa7Δ*tssm* against the effector deletion mutant LYSZa7Δ*tseL* and the effector-immunity deletion mutant LYSZa7Δ*tseL*Δ*tsiP*.

immunity protein-encoding genes (LYSZa7_12670 and LYSZa7_12675). The LYSZa7_12680 is predicted to encode a protein (WP_200200543.1) exhibiting both GxSxG and SDH catalytic motifs (Fig. 1A) (18). These two motifs are highly conserved in type VI lipase effector Tle1-4 in Gram-negative bacteria. In addition, it was predicted to be a homologue of the well-recognized lipase-effector TseL in *V. cholerae*, which belongs to the Tle2 subfamily of phospholipase. Further phylogenetic analysis revealed that LYSZa7_12680 encoded protein was highly identical to the predicted Tle2 in *P. aeruginosa* PA7 (19), with identity close to 96% (Fig. S1, Fig. 1B). Herein, we speculated that LYSZa7_12680 encoded a T6SS lipase effector, which belongs to the Tle2 family. Accordingly, we name LYSZa7_12680 as *tseL*, and its encoded protein WP_200200543.1 as TseL. Here, we use TseL[PA] to represent this newly identified TseL in *P. aeruginosa*.

To test if the TseL[PA] is a T6SS dependent effector, we constructed a deletion mutant, Δ*tseL*Δ*LYSZa7_12670*Δ*LYSZa7_12675* (Δ*tseL*Δ*tsiP* in short), lacking the predicted *tseL* functional region, including the toxin-encoding sequence and the immunity genes. By using LYSZa7 wild type and its T6SS defective mutant LYSZa7Δ*tssm* as killer strains, we compared the survival of Δ*tseL*Δ*tsiP* and Δ*tseL* after coincubation with the killers. The Δ*tseL*Δ*tsiP* was significantly outcompeted by wild type but not by the Δ*tssm*, and the survival of Δ*tseL* was slightly impacted after coincubation (Fig. 1C).

**TseL[PA] is a T6SS-dependent lipase effector with lipase-dependent antibacterial activity.** To examine the enzyme activity of TseL[PA], we purified TseL[PA] and performed the lipase-activity assay using Tween 20 as its catalytic substrate. Meanwhile, we mutated the catalytic motif of TseL[PA] by site-directed mutation of conserved amino acids (Fig. 2A) and purified the TseL[mutant] alleles for lipase-activity assay. The TseL[PA] showed significant catalytic activities, whereas the TseL[H466A] (site-directed mutation of histidine 466 to alanine) was unable to hydrolyze the substrate (Fig. 2B). Further, we showed that TseL[PA] was able to hydrolyze long-chain triglycerides by using olive oil and triolein as the substrates. The catalytic activity against long-chain triglycerides was compromised by the mutation of histidine 466 as well (Fig. S2).

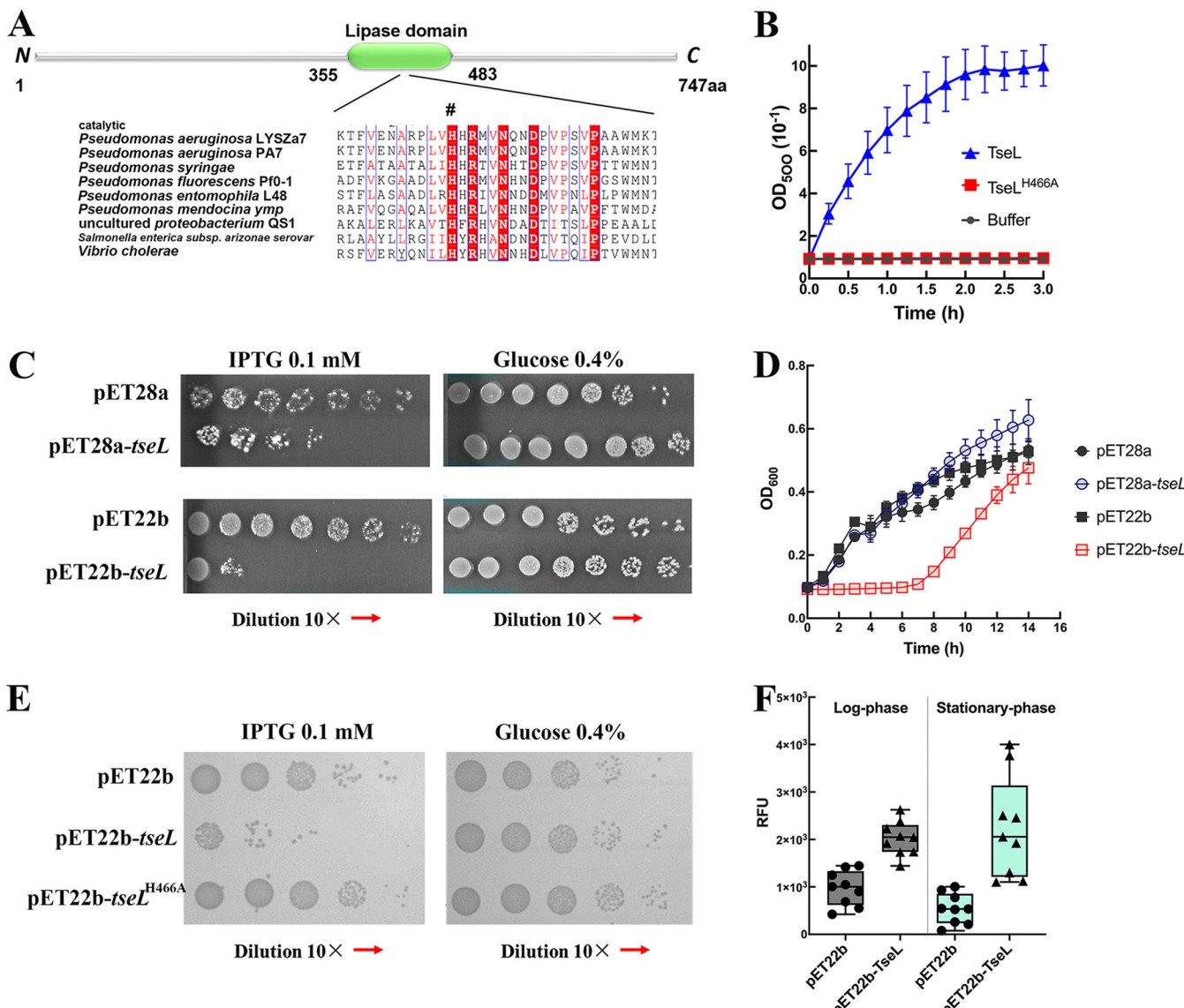

**FIG 2** (A) Prediction of TseL$^{PA}$ conserved domain and the conserved amino acids. (B) Lipase-activity of TseL$^{PA}$ and TseL$^{H466A}$ using Tween 20 as the substrate ($n$ = 3). (C) Bacterial toxicity assays of TseL$^{PA}$ when targeting periplasm (pET22b-*tseL*) or cytoplasm (pET28a-*tseL*). The TseL$^{PA}$ expression was induced by 0.1 mM IPTG, which was inhibited by 0.4% glucose. An empty vector was included as control for each condition. (D) Growth inhibition assay of TseL$^{PA}$ in *E. coli* BL21(DE3) by targeting periplasm or cytoplasm ($n$ = 3). (E) Bacterial toxicity assay of TseL$^{PA}$ and its site-directed mutant alle (TseL$^{H466A}$) in *E. coli* BL21(DE3). (F) Fluorescence detection of bacterial membrane damage ($n$ = 9). The degree of membrane damage was reflected by the intensity of red-fluorescence.

We next examined the lipase-dependent antibacterial activity of TseL$^{PA}$ by using bacterial toxicity assay. The induced production of TseL$^{PA}$ targeting periplasm (pET22b-*tseL*) rather than cytoplasm (pET28a-*tseL*) led to significant growth inhibition of *E. coli* (Fig. 2C and D). The expression of mutated TseL$^{PA}$ (TseL$^{H466A}$) failed to inhibit the growth of *E. coli* as expected (Fig. 2E). To address whether lipase-dependent antibacterial activity of TseL$^{PA}$ is attributed to the membrane damage, we examined the membrane integrity using the fluorescent probe, NPN (1-N-phenyl-naphtylamine). NPN is hydrophobic and cannot penetrate integrate bacterial membrane. Thereby, membrane damage can be reflected by the intracellular fluorescent intensity. We observed that NPN fluorescent intensity in *E. coli* expressing TseL was significantly higher than *E. coli* harboring the vector only. And this membrane damage effect was observed both in log phase and stationary phase (Fig. 2F).

Next, transcriptomic profiling of *E. coli* expressing TseL$^{PA}$ was analyzed in comparison with *E. coli* harboring vector to investigate the membrane-damage induced metabolic

dysregulations. Overall, there were 354 genes with significantly different expression levels (Fig. S3A, Table S3), among which 263 were downregulated and 91 were upregulated in *E. coli* BL21(DE3)::*tseL* (Fig. S3B). Function analysis of the dysregulated genes suggested that TseL$^{PA}$ boosts consumption of carbon and nitrogen sources and reduces iron acquisition, which are associated with bacterial stress responses (Fig. S3C-H). For example, the ABC transporter membranes for ferric compounds such as *fecCDER*, *fepBCDG,* and *fhuABCDE* were all downregulated in *E. coli* expressing *tseL*; the stress-response protein-encoding genes, including *yqgB*, *grcA,* and *ychH,* were dysregulated in the presence of TseL. In particular, the YidC-mediated membrane protein insertion pathway was found upregulated in the TseL-damaged *E. coli* cells, where *yidC* and *yidD* were 2-fold upregulated comparing to the control (Table S3). Furthermore, hexa-histidine-tag fused TseL$^{PA}$ was used to inspect the potential toxin targets in a pulldown assay. A total of 48 proteins were pulled out, most of which were small lipoproteins and membrane proteins (Table S4). In particular, the sensor protein Lrp, membrane protein YajC, and small heat shock proteins (sHsps, such as IpbA, IpbB, HslR, HslJ, etc.,) were identified. These potential TseL$^{PA}$ targeting proteins are involved in bacterial stress-shock responses, and the sHsps, specifically contribute to membrane stability.

**TsiP1 and TsiP2 are two immunity proteins of TseL$^{PA}$.** In the genomic region where TseL$^{PA}$ encoding gene was localized in LYSZa7 (Fig. 1), there are two open reading frames (ORFs) in the upstream of *tseL*, which are predicted to encode the immunity proteins for TseL$^{PA}$ (Fig. 1). Accordingly, we name the two ORFs *tsiP1* and *tsiP2*, respectively. The sequences of TsiP1 (WP_058160562.1) and TsiP2 (WP_023106862.1) are highly identical (92%), and both harbor the N-terminal signal peptides, suggesting their localization in periplasm (Fig. S4). To examine the immunity functions of TsiP1 and TsiP2, we coexpressed each of them with TseL$^{PA}$ in *E. coli* BL21(DE3) and examined the toxicity effect caused by TseL$^{PA}$. Both TsiP1 and TsiP2 can support the survival of *E. coli* from bacterial toxicity caused by TseL$^{PA}$, and the TsiP1 showed higher efficiency (Fig. 3A). Next, we examined the function of TsiP1 and TsiP2 in *P. aeruginosa* LYSZa7 using the bacterial growth competition assay. The *P. aeruginosa* LYSZa7Δ*tseL*Δ*tsiP1*Δ*tsiP2* knockout mutant failed in competing with the wild-type strain. Complementation of any of these two immunity proteins compensate for the growth disadvantage of LYSZa7Δ*tseL*Δ*tsiP1*Δ*tsiP2* against *P. aeruginosa* LYSZa7 (Fig. 3B).

To determine if the immunity proteins could physically interact with TseL$^{PA}$, we performed coimmunoprecipitation using *E. coli* where FLAG-tagged TseL$^{PA}$ was coexpressed with His-tagged immunity proteins. Western blot analysis confirmed that both immunity proteins can specifically bind to TseL$^{PA}$. The bottom panel of Fig. 3C shows a Western blot wherein TsiP1-His or TsiP2-His were incubated with anti-Flag beads in the absence of TseL-Flag. No binding was observed between TsiP-His and anti-Flag beads (Fig. 3C). Taken together, both TsiP1 and TsiP2 functionalize as immunities for their cognate toxin effector TseL$^{PA}$.

**TseL$^{PA}$ is widely distributed in clinical isolates.** To address whether this TseL$^{PA}$ is ubiquitous or is specially adopted in *P. aeruginosa* of certain linages, we blasted the sequence of TseL$^{PA}$ (WP_200200543.1) in the NCBI database. Phylogenetic analysis showed that TseL$^{PA}$ is commonly distributed across pathogens, including *P. aeruginosa*, *V. cholerae*, *Klebsiella pneumonia*, *E. coli*, etc. (Fig. 4). Further, we blasted the sequence of TseL against all isolates in the database of *Pseudomonas* Genome DB. There are 687 isolates that are able to encode the TseL$^{PA}$ protein (Table S5), among which 573 (83.4% out of total) are clinical isolates (Fig. S5). In addition, we screened 83 clinical isolates from *Pseudomonas* Genome DB by filtering out those with missing information in "host diseases." We found that 25.3% (21 out of 83) of the examined strains encode TseL$^{PA}$ protein, including the frequently picked model organisms for pathogenicity studies such as PA7, PAK, and PA1 (Fig. S6). Therefore, we speculated that the acquirement of TseL might contribute to the interbacterial competition and host virulence of *P. aeruginosa* and lead to emergence of new epidemic clonal linages.

**TseL$^{PA}$ contributes to both intra- and interspecies competitions.** To examine whether TseL$^{PA}$ contributes to interbacterial competitions, we carried out inter- and intraspecies

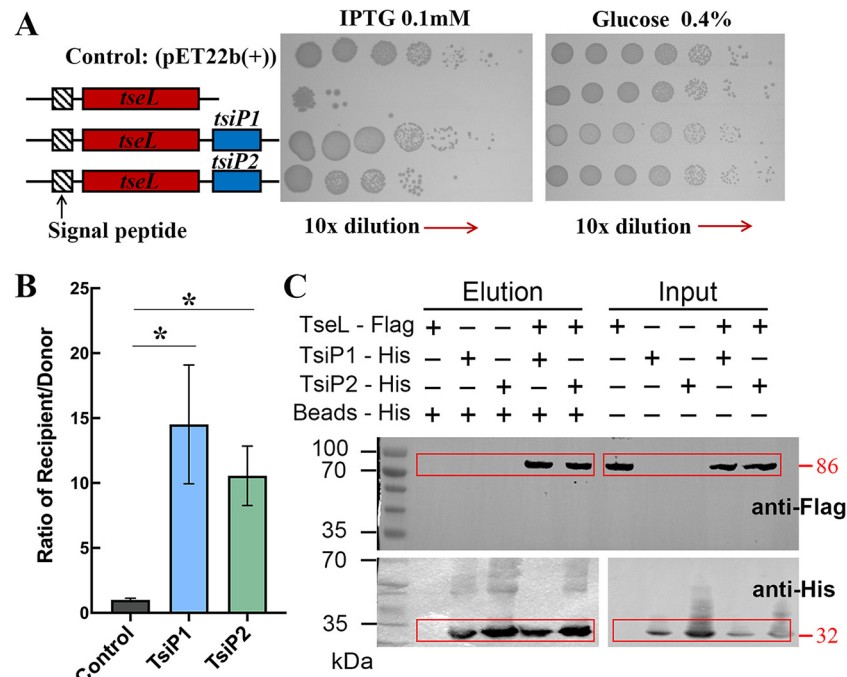

**FIG 3** (A) Bacterial toxicity assay of *E. coli* expressing TseL[PA], TseL[PA]-TsiP1, and TseL[PA]-TsiP2, respectively. (B) Competition assay of LYSZa7 and its mutants. The donor strain is *P. aeruginosa* LYSZa7 wild type across the experiment. The recipient strain is the knockout mutant LYSZa7Δ*tseL*Δ*tsiP1*Δ*tsiP2* with the TsiP1/TsiP2 complemented strain, respectively ($n = 4$; *, $P < 0.05$). (C) Coimmunoprecipitation of TseL and TsiP1/TsiP2.

competition assays. First, we inspected the role of TseL[PA] in intraspecies competition. By comparing the growth competition advantages between LYSZa7 and LYSZa7Δ*tseL* against different recipients, we found TseL[PA] was important when competing with PA14 and PAO1 that do not encode TseL[PA] or its immunities (Fig. 5A). No difference was observed in the fitness competition between LYSZa7 wild type and TseL[PA] knockout strain when competing with PAK which also expresses TseL[PA]. Interestingly, in the competition assay against *P. aeruginosa* PDO300, an isogenic mucoid strain of PAO1 without the TseL[PA]-encoding gene, the effect of TseL[PA] was imperceptible perhaps due to the protection effect of its thick alginate exopolysaccharides. Next, we examine the contribution of TseL[PA] in interspecies competition by using *E. coli* BL21, *Acinetobacter baumannii* ATCC17978, and *K. pneumoniae* as recipients (Fig. 5B). The expression of TseL[PA] in LYSZa7 improved the growth competition capability against *E. coli* and *A. baumannii*, but not *K. pneumonia*.

**TseL contributes to host invasion processes of *P. aeruginosa*.** In order to investigate whether *tseL* enhances the virulence of *P. aeruginosa* to eukaryotic cells, human alveolar epithelial cell A549 of lung cancer and HeLa cell line were selected for bacterial infection experiment. The deletion of *tseL* in LYSZa7 reduced its invasion capability toward HeLa and A549 up to 4-fold (Fig. 6). Further, we inspected the phagocytosis resistance of LYSZa7 and its TseL[PA] mutant by Raw 254.7 macrophage by counting the internalized bacteria. An approximately 10-fold difference was observed between wild type and the mutant, indicating strong antiphagocytic potency by expressing TseL in LYSZa (Fig. 6).

## DISCUSSION

Bacterial T6SS is widely distributed among Gram-negative bacteria, which provides competitive advantage and contributes to bacterial pathogenesis as a contact-based secretory apparatus. More recently, the T6SS has been shown to contribute to host colonization, nutrient uptake, biofilm formation, and quorum sensing (QS) related activities (19, 21). The functions of T6SS mainly rely on its effectors, which target various essential cellular

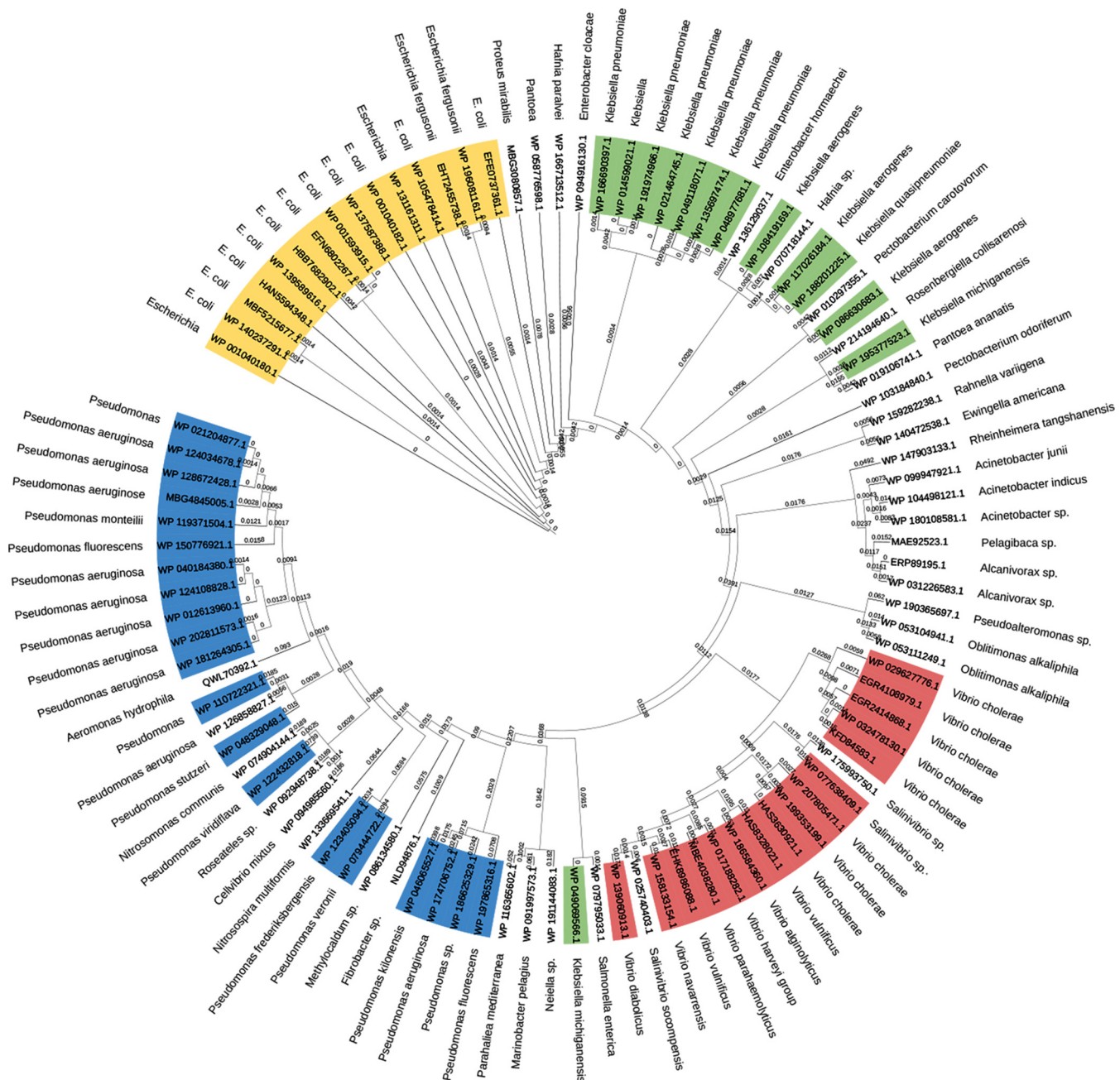

**FIG 4** A. Phylogenetic analysis of the top 90 hits of TseL[PA] in NCBI database.

components, such as the cell membrane, cell wall, and DNA. Numerous T6SS-dependent effectors have been predicted by comparative genomic analysis in a broad range of bacteria (13). The characterization of T6SS effectors of *P. aeruginosa*, the most predominant opportunistic pathogens in nosocomial infections (20, 22), is mainly focused on certain model strains such as PAO1 and PA14 (23, 24).

In this work, we experimentally characterized a TseL homologue and its immunity proteins TsiP1 and TsiP2 in a *P. aeruginosa* clinical isolate in a COVID-19 patient. We proved that this TseL[PA] belongs to the Tle2 family, with high identity to the putative Tle2 in *P. aeruginosa* PA7 (Tle2[PA7]) predicted in a previous study by bioinformatic analysis (18, 25). Although two immune proteins of Tle2[PA7] were predicted, no additive effect of protection was observed by harboring the two immunity proteins simultaneously, which required further inspection.

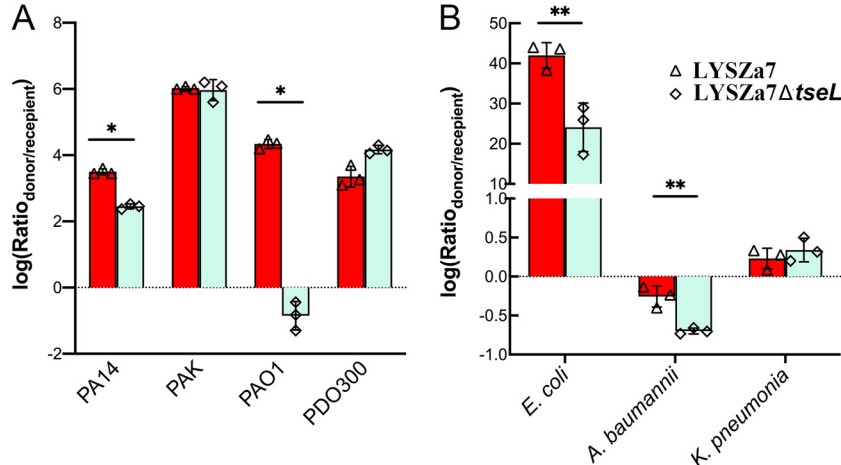

**FIG 5** Intra- and inter-species competition assays by *P. aeruginosa* LYSZa7 and *P. aeruginosa* LYSZa7Δ*tseL*. (A) Intraspecies competition by using *P. aeruginosa* LYSZa7 and *P. aeruginosa* LYSZa7Δ*tseL* as donor strain, and PA14, PAK, PAO1 and PDO300 as recipients, respectively. (B) Interspecies competition by using *P. aeruginosa* LYSZa7 and *P. aeruginosa* LYSZa7Δ*tseL* as donor strain, and *E. coli* BL21, *A. baumannii* ATCC17978, and *K. pneumonia* BAA1705 as recipients, respectively. ($n = 3$; *, $P < 0.05$; **, $P < 0.01$).

The activity of effectors is crucial for T6SS functions. The TseL$^{PA}$ shows lipase-dependent antibacterial activity, which enhanced LYSZa7 competition against *P. aeruginosa* strain that does not harbor this TseL$^{PA}$ homologue. Also, TseL$^{PA}$ contributes to the interspecies competition of *P. aeruginosa* against *E. coli* BL21(DE3) and *A. baumannii* ATCC17978. Interestingly, no advantages are observed in LYSZa7 harboring TseL when it competes with *P. aeruginosa* PDO300, a mucoid variant, and with *K. pneumonia* BAA1705. Since both *P. aeruginosa* PDO300 and *K. pneumonia* are highly active in exopolysaccharide production, it is speculated that antibacterial activity of TseL$^{PA}$ is compromised by the presence of these surface protecting heavy molecule-weight components (26).

Eukaryotic targeting T6SS effectors usually involve diverse cellular processes, such as adhesion modification, stimulating internalization, cytoskeletal rearrangements, and evasion of host innate immune responses, to allow pathogens to colonize, survive, and disseminate in host environment (27). A recent study reported that clinical isolates of *Pa* showed reduced virulence with attenuated T6SS functions during short-term colonization in COVID-19 patients, which was proposed to help bacteria escape the immune elimination effect of the host (28). In this work, TseL$^{PA}$ is found to render *Pa* higher virulence and anti-phagocytosis against host cells. Further studies are required to answer

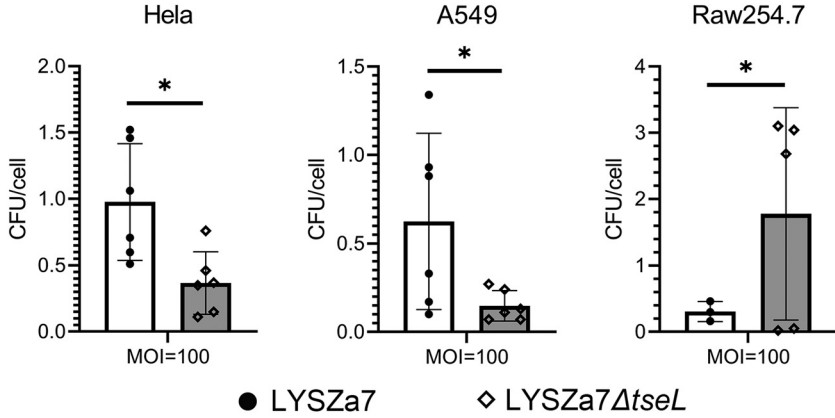

**FIG 6** Invasion assay/Bacterial internalization assay of *P. aeruginosa* LYSZa7 and *P. aeruginosa* LYSZa7Δ*tseL* to HeLa, A549 and Raw254.7 cells. The internalized cells were calculated and normalized based on the number of internalized bacteria per mammalian cell. ($n = 6$; *, $P < 0.05$).

the question that what benefits could be rewarded by expression and assembly of TseL[PA] to overcome the host immune system.

Furthermore, the *tseL*[PA] is found adjacent to a VgrG encoding gene to form a *vgrG-tseL* gene cluster that is observed primarily in clinical isolates instead of model organism PAO1 or environmental isolates (Table S5, Fig. S5). The transmission of T6SS effectors has been reported for many Tle family effectors. *Pa* was also proposed in a previous study to acquire its eukaryotic-like phospholipase D (H2-T6SS) by horizontal acquisition (29). Although *Pa* has a nonclonal epidemic population structure, hospital transmission is able to raise the prevalence of certain adapted clones (30, 31). Acquisition of the accessory effector TseL[PA] enhanced *Pa*'s fitness over the same species and enriched the population diversity in the infection sites. It is speculated that this *vgrG-tseL*[PA] gene cluster is acquired by LYSZa7 via horizontal gene transfer during the environmental adaptation specifically in hospitals.

**Conclusion.** In this work, we discovered and characterized a T6SS-dependent lipase effector, TseL, and its immunity proteins TsiP1 and TsiP2 widely distributed in the *P. aeruginosa* clinical strains. The TseL in *P. aeruginosa* (TseL[PA]) carries a conserved catalytic triad (S-H-D) and a GxSxG motif, as a homologue of the TseL in *V. cholerae* (TseL[Vc]), which belongs to the Tle2 subfamily of phospholipase. Our findings provide experimental evidence of the conserved H466 histidine residue in the domain that is required for the lipase activity of TseL[PA] and the correspondence antibacterial toxicities. Also, we demonstrated that TseL[PA] contributes to *P. aeruginosa*'s interbacterial competition and host pathogenesis. And the TseL homologues exist in diverse species, pathogens in particular, and are commonly associated with the upstream VgrG encoding gene. Collectively, this work demonstrates that TseL represents an important member of cell membrane-targeting T6SS effector targeting both prokaryote and eukaryote and might play a role in emergence of successful *P. aeruginosa* epidemic linages.

## MATERIALS AND METHODS

**Bacterial strains, plasmids, cell lines, and growth conditions.** All bacterial strains and plasmids used in this study were listed in Table S1. Unless otherwise stated, *P. aeruginosa*, *E. coli*, *A. baumannii*, and *K. pneumonia* strains were cultivated at 37°C in LB broth. Antibiotics were supplemented to the medium when necessary: gentamicin (Gm) 60 $\mu$g/mL, chloramphenicol (Cm) 6 $\mu$g/mL, irgasan (Irg) 25 $\mu$g/mL. *P. aeruginosa* LYSZa7 and *E. coli* BL21(DE3) were used as model organisms throughout this work. The in-frame deletion of LYSZa7 was performed by modified two-step allelic exchange as reported previously (32). Details are described in the Supplementary Methods, and all primers used in this work are listed in Table S2. Plasmids pET28 a(+) and pET22b (+) were used for inducible expression of target genes in *E. coli* BL21(DE3), and plasmid MiniCTX-1 were used for constitutive expression of target genes in *E. coli* BL21(DE3) and *P. aeruginosa* LYSZa7 (33).

**Bacterial toxicity assay.** *E. coli* BL21(DE3) pET28a-*tseL* with inducible TseL targeting in cytosol and *E. coli* BL21(DE3) pET22b-*tseL* with inducible TseL targeting in periplasm were cultivated overnight followed by serial diluted with dilution factor at 10 in LB broth. The diluted bacterial suspension was spotted with 5 $\mu$L each onto LB agar plate containing 0.1 mM IPTG or 0.4% glucose. CFU were counted and analyzed after 24 h static cultivation. Growth of BL21(DE3) pET28a-*tseL* and BL21(DE3) pET22b-*tseL* were evaluated in liquid cultures. Similarly, toxicity evaluation of *E. coli* BL21(DE3) expressing *tseL-tsiP1/tsiP2* loci was conducted by the same method.

**Fluorescence detection of bacterial membrane integrity.** In this experiment, the GENMED n-phenylnaphthalamine uptake kit (GMS60045.2 v.A) was used to detect the degree of bacterial membrane damage. Overnight bacterial solution was transferred to fresh liquid LB (1:100 *v:v*) and was incubated at 37°C and 200 rpm. Then, the fresh bacterial culture (OD$_{600}$ = 0.50 or plateau stage) was centrifuged for 1 min at 16,000 g (or 13,000 rpm). Following the manufacturer's instruction, the reaction system was incubated at 25°C for 5 min and the fluorescence was detected with a fluorimeter (TECAN ENCO) immediately (Ex/Em: 355/460 nm).

**Bacterial competition assay.** Intra- and interspecies competition assays were performed as described previously (29). The recipient strain LYSZa7 contained either chromosomal integrated or plasmid conferred tetracycline resistant gene for selection. The other recipient strains contained plasmid pBBR1MCS5 conferred gentamicin resistant gene for selection. Donor and recipient strains were grown overnight and washed with sterile ddH$_2$O, before being mixed at the appropriate ratio in ddH$_2$O. The mixture was spotted onto pieces of sterile nitrocellulose membrane (Lot 082821210831, Beyotime, China) overlaid onto a 3% LB low salt (LB-LS) agar plate (LB low salt: 10 g peptone and 5 g yeast extract per L). The plates were incubated at 37°C for 48 h. Colonies were scraped off and suspended in 1 mL LB medium, followed by the serial dilution with a diluting factor at 10. The bacterial suspensions were spotted onto both nonselective and Gem-selective LB agar plates with 5 $\mu$L each. CFU were counted and the difference between the donor-recipient ratios was determined. For *P. aeruginosa* intraspecies competition, the ratio of donor to recipient was set at 5, and the incubation

duration was 48 h at 37°C For bacterial interspecies competition, the donor-recipient ratio was 1:1 and the incubation duration was 24 h at 37°C.

**Bacterial infection assay.** HeLa cells, human alveolar epithelial cell A549 and macrophage RAW254.7 were used for bacterial infection in this study. Cells were cultured in RPMI 1640 (Roswell Park Memorial Institute 1640, Sangon Biotech) supplemented with 10% fetal bovine serum (FBS) in 24-well plates. When grown to 75% confluence at 37°C (under 5% $CO_2$), cells were washed twice with sterile PBS buffer, and were infected with *P. aeruginosa* LYSZa7 or its mutants from stationary phase at an MOI of 50 or 100 for 2 h. Subsequently, the infected cell cultures were washed twice with PBS buffer before being incubated in RPMI 1640 containing 200 $\mu$g/mL gentamicin at 37°C for 2 h. Thereafter, the cells were washed with PBS three times before lysis with ddH2O containing 0.1% Triton X-100 on ice for 30 min. CFU/ were counted to determine the number of internalized bacteria in the cells (29).

**Protein purification and enzyme activity.** Coding sequence of *tseL* from LYSZa7 was inserted into pET28a-SUMO vector. Overexpression of TseL was induced in *E. coli* BL21(DE3) pLysSpET28-SUMO-*tseL* by 0.5 mM IPTG for overnight at 18°C. Cells were harvested and resuspended in binding buffer (50 mM Tris, pH 8.0, 500 mM NaCl, and 10 mM imidazole). Bacteria were lysed using probe sonication for 40 min, and cell debris were removed by centrifugation at 15,000 rpm for 30 min. The supernatant was loaded to a Ni-NTA column (L00250-50, GeneScript). Proteins were eluted with elution buffer (50 mM Tris, pH 8.0, 500 mM NaCl, and 300 mM imidazole). Sumo enzyme (Ulp) was added to protein solution, and the mixture was incubated at 4°C overnight, followed by centrifugation at 5,000 rpm for 5 min. The supernatant was collected and loaded onto a Ni-NTA column with target proteins being eluted by the elution buffer. The purified proteins were concentrated for downstream experiment.

The lipase activity of TseL and its site-directed mutants were determined using Tween 20, olive oil, or triolein as the substrates (34). Activities of proteins on each substrate were measured using 100 $\mu$g purified enzymes unless otherwise indicated. The enzyme activity using Tween 20 as a substrate was assayed at 37°C in a buffer containing 20 mM Tris, pH 8.0, 2% (vol/vol) Tween 20, and 3 mM $CaCl_2$ (35). The optical density at 500 nm was monitored. The enzyme activity using olive oil or triolein as the substrates was monitored by using rhodamine B (0.001% wt/vol) as indicator, and the fluorescence reading was determined with Ex/Em at 350 nm/580 nm.

**Coimmunoprecipitation and Western-blot analysis.** For immunoprecipitation assays, *tseL* was cloned into pME6032 with 3′ fusion of a FLAG epitope tag, and *tsiP1* and/or *tsiP2* were cloned into pBAD/Myc-His A to generate 3′ Myc-His tag fused expressing plasmids. FLAG-tagged and His-tagged proteins were coexpressed in BL21(DE3) (DE3) pLysS and were induced by 0.5 mM IPTG and 0.2% arabinose for 3 h at 37°C when $OD_{600}$ reaches 0.4. Bacteria were harvested by centrifugation at 8000 g for 5 min and were resuspended in IP-lysis buffer (50 mM Tris, pH 7.5, 250 mM NaCl, and 6 mM KCl) containing a protease inhibitor cocktail (Roche). Cells were lysed by probe sonication and cell debris were removed by centrifugation at 15,000 rpm for 20 min at 4°C. Supernatants were passed through a 0.22 $\mu$m filter and were incubated with magnetic beads overnight. Tagged proteins were eluted by 1× SDS-PAGE loading buffer. For Western blot, protein samples were separated by SDS-PAGE and transferred to PVDF membrane (Millipore) using a wet blotter (Bio-Rad, Mini Trans-Blot Cell). Membranes were probed using a standard protocol. In brief, the membrane was blocked in TBST containing 5% skimmed milk at room temperature for 1 h and incubated with primary antibody overnight at 4°C. Subsequently the membrane was washed five times with TBST (1×) before incubation with a secondary antibody for 1h at room temperature. After a further three washes in TBST, the immune reactive proteins could be localized. The primary antibodies were polyclonal rabbit anti-His and anti-FLAG antibodies (Sigma) used at a dilution of 1:1000. The secondary antibody used was an anti-mouse IgG1 secondary antibody and used at a dilution of 1:5000. Immune reactive bands were visualized by Chemiluminescence imaging system (ChemiScope-6100PLUS).

**RNA extraction and purification.** The bacterial cells were first treated with RNA Protect Reagent (Qiagen, Germany) to maintain the integrity of RNA. The total RNA was extracted from these bacterial cells using a miRNeasy minikit (Qiagen, Germany) with modifications. A Turbo DNA-free kit (Thermo Fisher Scientific, Lithuania) was used to remove genomic DNA contaminants from total RNA. DNA contamination was assessed with a Qubit dsDNA High Sensitivity assay (PicoGreen dye) and a Qubit 2.0 Fluorometer (Invitrogen, Austria) according to manufacturer's instructions. rRNA was depleted with a Ribo-Zero rRNA removal kit (Illumina, USA). The integrity of the total RNA was assessed with an Agilent TapeStation System (Agilent Technologies, UK). The double-stranded complementary DNAs (cDNAs) were reverse-transcribed using a NEBNext RNA first and second strand synthesis module (NEB, USA). cDNAs were subjected to Illumina's TruSeq Stranded mRNA protocol. The quantitated libraries were then pooled at equimolar concentrations and sequenced on an Illumina HiSeq2500 sequencer in rapid mode at a read-length of 100 bp paired-ends.

**Transcriptomic analysis.** The *E. coli* str. K-12 substr. MG1655 genome (NC_000913) was used as the reference for *E. coli* transcriptomics analysis. The RNA-Seq raw data were analyzed using the "RNA-Seq and expression analysis" application in CLC genomics Workbench 10.0 (Qiagen). The total gene reads from CLC genomics Workbench 10.0 were subjected to the DESeq2 package for statistical analysis68 by using R/Bioconductor.69 A hierarchical clustering analysis was performed with a negative binomial test using the DESeq2 package. The heatmap.2 package was used to draw a heat map for the differentially expressed genes of *E. coli* cells with fold change larger than two and an adjusted *P* value smaller than 0.05. The principal-component analysis plot was generated in R/Bioconductor. The adapter-trimmed and assembled RNA sequences of the mixed-species microbial community were used as metatranscriptomic data for analysis. The output aligned gene reads in DAA format were uploaded and "meganized" in MEGAN6.11.1 with a minimum bit-score of 50 and a top percentage of 25. Functional analyses of these aligned gene reads were performed using SEED classifications. A PCoA was plotted to cluster the samples based on functions. The functional analysis results are illustrated in stacked bar charts.

**Data availability.** All Illumina sequencing data used in this study could be found under BioProject No. PRJNA826202.

## SUPPLEMENTAL MATERIAL

Supplemental material is available online only.

**SUPPLEMENTAL FILE 1**, PDF file, 1.8 MB.

## ACKNOWLEDGMENTS

This work was supported by the Guangdong Natural Science Foundation for Distinguished Young Scholar (2020B1515020003); the Guangdong Basic and Applied Basic Research Foundation (2019A1515110640); the Guangdong Basic and Applied Basic Research Foundation (2020A1515010316); the Shenzhen Key Laboratory of Gene Regulation and Systems Biology, Southern University of Science and Technology (ZDSYS20200811144002008); and the Shenzhen Science and Technology Program (KQTD20200909113758004).

The authors have no conflicts of interest to declare.

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
