## [Reviewer comments · Microbiology Spectrum]

Microbiology Spectrum

Acquisition of T6SS effector TseL contributes to the emerging of novel epidemic strains of *Pseudomonas aeruginosa*

Anmin Ren, Minlu Jia, Ji-Hong Liu, Tian Zhou, Liwen Wu, Tao Dong, Zhao Cai, Jiu-Xin Qu, Yang Liu, Liang Yang, and Yingdan Zhang

Corresponding Author(s): Yingdan Zhang, School of Medicine, Southern University of Science and Technology

Review Timeline:

Submission Date:	August 20, 2022
Editorial Decision:	October 4, 2022
Revision Received:	November 14, 2022
Accepted:	November 16, 2022

Editor: Beile Gao

Reviewer(s): The reviewers have opted to remain anonymous.

Transaction Report:

DOI: <https://doi.org/10.1128/spectrum.03308-22>

October 4, 2022

Dr. Yingdan Zhang
School of Medicine, Southern University of Science and Technology
Shenzhen
China

Re: Spectrum03308-22 (Acquisition of T6SS effector shapes the evolution of novel epidemic strains of *Pseudomonas aeruginosa*)

Dear Dr. Yingdan Zhang:

Link Not Available

Sincerely,

Beile Gao

Journals Department
Reviewer comments:

Reviewer #1 (Comments for the Author):

In this paper, Ren, Jia et al. describe the functional characterization of a toxin/immunity pair found in a VgrG/T6SS cluster of a *Pseudomonas aeruginosa* clinical strain (LYSza7). The authors provide biochemical and genetics evidence that the TseL toxin acts as a phospholipase, which is important to intra- and interspecies competition and to *P. aeruginosa* invade host cells. The manuscript is well presented and written, the experiments seem to have been well-executed, and the data are correctly described and analyzed. However, some questions need to be addressed.

Major concerns:

1) The authors stated that TseL is a T6SS-dependent effector (Lines 1, 20, 56, 114, 136). However, there is no experimental evidence indicating that TseL is a T6SS substrate. The authors must perform competition assays using mutant strains defective

- to T6SS machinery genes to confirm the dependence of T6SS to TseL secretion;
- 2) The authors emphasize aspects related to the evolution of *P. aeruginosa* clinical strains that are not supported by their data (Lines 1, 20, 42, 43, 58, 59, 235, 236). These statements should be restricted to the Discussion section as speculation. The title and running title must be modified to reflect the main author's experimental findings;
 - 3) Several sentences are missing references (Lines 123, 283, 288, 294). The reference list should be more comprehensive, including more works describing T6SS effectors of *P. aeruginosa*.
 - 4) The Figures in the PDF file show low quality.

Minor concerns:

- 1) Line 25: others what?
- 2) Line 29: "in clinical evolution of *P. aeruginosa*"... must be "in evolution of clinical *P. aeruginosa*";
- 3) Lines 64 and others: Gram instead gram;
- 4) Line 74: "applies multiple approaches". This sentence must be rephrased.
- 5) Line 83: Change "although" by "However";
- 6) Line 87: The sentence must be rephrased;
- 7) Line 106: Change the word "unillustrated";
- 8) Line 129: Use "Tle2 family" instead of "family of Tle2";
- 9) Figures 1A and 2A: The authors should standardize the scheme of TseL (lipase domain is the entire protein as in 1A or only the central portion as in 2A);
- 10) Lines 187 and 188: The sentence must be rephrased;
- 11) Line 193: Remove "As shown";
- 12) Line 194: Move (Figure 1) to the end of the sentence;
- 13) Line 196: Correct "ORFs";
- 14) Why the authors choose 95% identity?
- 15) Lines 254-256 and 263: It is missing to describe *E. coli* competition;
- 16) Line 265: There is no data about toxicity in figure 6;
- 17) Lines 272-273: The sentence must be rephrased;
- 18) Lines 281-282: The sentence must be rephrased;
- 19) Line 314: There is no data about autophagy in the paper;
- 20) Line 319: Did the authors analyzed environmental strains?

Reviewer #2 (Comments for the Author):

In the present work, Ren et al has nicely revealed a type VI secretion system (T6SS) dependent lipase effector in *P. aeruginosa* LYSZa7, which is a homologue of TseL in *Vibrio cholerae* and is widely distributed in pathogens. The authors validated that this TseL homologue belongs to the Tle2, and named this effector TseLPA. The toxicity of TseLPA can be neutralized by two immunity proteins TsiP1 and TsiP2 which are encoded upstream of tseL. TseLPA contributes to bacterial pathogenesis by promoting bacterial internalization into host cells. The reviewer has the following comments to improve the manuscript.

1. Some pictures are too small to be seen clearly.
2. Missing lane labelling for Figure 3C.
3. Some typos in the manuscript, such as "Echerichia" in Figure 4A.
4. Suggest remove "*P. aeruginosa*" the subtitle "TseLPA is widely distributed in *P. aeruginosa* clinical isolates".
5. TseLPA may be a bit misleading since only a small part of *P. aeruginosa* strains contain it.

Staff Comments:

Preparing Revision Guidelines

- Point-by-point responses to the issues raised by the reviewers in a file named "Response to Reviewers," NOT IN YOUR COVER LETTER.
- Upload a compare copy of the manuscript (without figures) as a "Marked-Up Manuscript" file.

- Each figure must be uploaded as a separate file, and any multipanel figures must be assembled into one file.
- Manuscript: A .DOC version of the revised manuscript
- Figures: Editable, high-resolution, individual figure files are required at revision, TIFF or EPS files are preferred

Please return the manuscript within 60 days; if you cannot complete the modification within this time period, please contact me. If you do not wish to modify the manuscript and prefer to submit it to another journal, please notify me of your decision immediately so that the manuscript may be formally withdrawn from consideration by Microbiology Spectrum.

November 16, 2022

Dr. Yingdan Zhang
School of Medicine, Southern University of Science and Technology
Shenzhen
China

Re: Spectrum03308-22R1 (Acquisition of T6SS effector TseL contributes to the emerging of novel epidemic strains of *Pseudomonas aeruginosa*)

Dear Dr. Yingdan Zhang:

Your manuscript has been accepted, and I am forwarding it to the ASM Journals Department for publication. You will be notified when your proofs are ready to be viewed.

Sincerely,

Beile Gao
Editor, Microbiology Spectrum
